

# Embryo and larval biology of the deep-sea octocoral *Dentomuricea* aff. *meteor* under different temperature regimes

Maria Rakka[1,2]   António Godinho[1,2]   Covadonga Orejas[3]   Marina Carreiro-Silva[1,2]

[1] IMAR-Instituto do Mar, Universidade dos Açores, Horta, Portugal
[2] Okeanos-Instituto de Investigação em Ciências do Mar da Universidade dos Açores, Horta, Portugal
[3] Centro Oceanográfico de Gijón, Instituto Español de Oceanografia, IEO, CSIC, Gijón, Spain

## ABSTRACT

Deep-sea octocorals are common habitat-formers in deep-sea ecosystems, however, our knowledge on their early life history stages is extremely limited. The present study focuses on the early life history of the species *Dentomuricea* aff. *meteor*, a common deep-sea octocoral in the Azores. The objective was to describe the embryo and larval biology of the target species under two temperature regimes, corresponding to the minimum and maximum temperatures in its natural environment during the spawning season. At temperature of 13 ±0.5 °C, embryos of the species reached the planula stage after 96h and displayed a median survival of 11 days. Planulae displayed swimming only after stimulation, swimming speed was 0.24 ±0.16 mm s$^{-1}$ and increased slightly but significantly with time. Under a higher temperature (15 °C ±0.5 °C) embryos reached the planula stage 24 h earlier (after 72 h), displayed a median survival of 16 days and had significantly higher swimming speed (0.3 ±0.27 mm s$^{-1}$). Although the differences in survival were not statistically significant, our results highlight how small changes in temperature can affect embryo and larval characteristics with potential cascading effects in larval dispersal and success. In both temperatures, settlement rates were low and metamorphosis occurred even without settlement. Such information is rarely available for deep-sea corals, although essential to achieve a better understanding of dispersal, connectivity and biogeographical patterns of benthic species.

## INTRODUCTION

Species persistence requires the successful completion of a life cycle against biotic and abiotic odds, in most cases starting with survival at early life history stages. For benthic marine invertebrates, larval stages constitute the only pelagic phase that ensures dispersal and connectivity among populations (*Cowen & Sponaugle, 2009*). Moreover, early life events such as larval survival and settlement determine the fate of the sessile, adult phase and are extremely important (*Marshall & Morgan, 2011*; *Byrne, 2012*). In deep-sea communities, which are dominated by benthic marine invertebrates, knowledge on early life stages is therefore key in understanding species distributions, biogeographical patterns

Corresponding author
Maria Rakka,
marianinha.rk@gmail.com

and metapopulation dynamics (*Treml et al., 2015*), constituting an essential tool for management (*Hilário et al., 2015*).

Deep-sea octocorals are major habitat-formers in the deep-sea, usually occurring in complex geological settings such as continental shelves and margins (*Yesson et al., 2012*; *Taylor et al., 2013*), underwater canyons (*Brooke et al., 2017*) and seamounts (*Tempera et al., 2012*; *Braga-Henriques et al., 2013*). Due to the habitat requirements of some octocoral species, including hard substrates for settlement and strong currents which optimize food delivery, their distribution can be quite patchy (*Bryan & Metaxas, 2006*; *Tong et al., 2012*), as observed for other deep-sea benthic species (*Miller & Gunasekera, 2017*). Anthropogenic disturbance and global climate change are likely to cause habitat fragmentation by altering its characteristics (*Sweetman et al., 2017*; *Levin et al., 2019*) and causing a decrease in the available suitable habitat of some species (*Morato et al., 2020*). Under these circumstances, obtaining a solid understanding of larval biology and population connectivity is essential to understand community dynamics and the potential of deep-sea octocoral populations to recover from disturbance (*Cowen et al., 2007*; *Levin et al., 2020*).

So far, our knowledge on larval biology of deep-sea octocorals is limited to a few brooding species (*Cordes, Nybakken & Van Dykhuizen, 2001*; *Sun, Hamel & Mercier, 2010*; *Sun, Hamel & Mercier, 2011*; *Mercier & Hamel, 2011*). In most of these cases, larvae displayed short competency periods with limited swimming behaviour (*Sun, Hamel & Mercier, 2010*), settlement within 2-5 days after release and rapid metamorphosis into primary polyps (*Cordes, Nybakken & Van Dykhuizen, 2001*; *Sun, Hamel & Mercier, 2011*). However, many deep-sea octocorals are broadcast-spawners and are therefore expected to display different larval characteristics and dispersal capabilities (*Harrison & Wallace, 1990*; *Nishikawa, Katoh & Sakai, 2003*). To our knowledge, up to date there is no detailed description of embryo and larval development of broadcast spawning deep-sea octocorals. Larvae from broadcast spawning species undergo early development in the water column, where they are mostly transported as passive particles until they reach the planula stage. During transportation, embryos can be exposed to variable environmental conditions which may affect their development (*Melzner et al., 2009*). This phenomenon can be even more pronounced in larvae of deep-sea species, which often display upward swimming, crossing water masses with very different physicochemical characteristics (*Young et al., 1996*; *Young et al., 2012*; *Arellano et al., 2014*; *Strömberg & Larsson, 2017*). In the case of deep-sea corals, the effect of natural fluctuations of environmental conditions, such as salinity and temperature, have only been addressed in the scleractinian *Lophelia pertusa* (Stromberg & Larsson, 2017).

The aim of this study was to provide a detailed description of the early life history traits of the deep-sea broadcast spawning species *Dentomuricea* aff. *meteor,* a common habitat-forming, deep-sea octocoral in the Azores. More specifically objectives were (1) to describe the embryo and larval development, larval survival, swimming and settlement behaviour of the target species and (2) to determine the effect of natural temperature variability on its embryo and larval traits. To achieve these objectives, we employed an experimental approach with assisted fertilization and larvae rearing in aquaria under

two temperature regimes (13 ±0.5 °C and 15 ±0.5 °C), representing the minimum and maximum temperatures experienced by the species in its natural habitat.

## MATERIALS AND METHODS

### Target species and colony collection

The Azores Archipelago, located above the Mid-Atlantic Ridge, is a biodiversity hotspot for deep-sea octocorals (*Sampaio et al., 2019*). Coral gardens (*OSPAR, 2010*) formed by deep-sea octocorals are among the most prominent deep-sea communities on regional seamounts and island slopes (*Braga-Henriques et al., 2013*). *Dentomuricea* aff. *meteor* is an octocoral species of the family Plexauridae, so far only recorded on the seamounts of the North Mid-Atlantic Ridge. It is common in regional seamounts between 200–600 m (*Braga-Henriques et al., 2013*), where it forms dense populations, often in combination with other octocoral species such as *Viminella flagellum* and *Callogorgia verticillata* (*Tempera et al., 2012*). The species is gonochoristic and presents gametes all year round, with seasonal peaks of gamete maturation and spawning usually occurring in autumn (M Rakka, 2020, unpublished data).

A total of 11 colonies of the species *Dentomuricea* aff. *meteor* were collected as by-catch from experimental long-line fisheries on board RV Archipelago (ARQDAÇO monitoring programme). Collection was performed at the summit of Condor Seamount, between 200–280 m, in September and October 2019. Colonies were divided in large fragments (20–30 cm height) and were kept at the DeepSeaLab aquaria facilities (*Orejas et al., 2019*), in six 33L aquaria positioned in a thermo-regulated room at 14 °C. Aquaria were supplied continuously with seawater (SW) pumped from 5m depth, previously treated with UV light (P10 UVsystem & Vecton 600 TMC$^{TM}$) and passed through 50 µm and 1 µm mesh filters. Circulation within the aquaria was maintained by pumps. Seawater temperature was kept between 13–14 °C with the aid of chillers and salinity was 35.8 ±0.1, similar to the natural conditions at the collection site (*Santos et al., 2013*). Colonies were fed twice per day with a mixture of frozen zooplankton and microplankton which was frequently enriched with live microalgae (*Chaetoceros calcitrans* and *Nannochloropsis gaditana*) and live rotifers.

### Larval rearing

Larvae were obtained by maintaining reproductively active female and male colonies in the same aquaria to achieve natural spawning and fertilization. Coral fragments were allowed to acclimatize in the above aquaria conditions for approximately one month. Subsequently, colonies with mature gametes were identified by dissecting two branchlets (3–5 cm height) from each colony and observing their tissue under a dissecting microscope. Reproductively immature colonies and fragments in poor condition were excluded from further analysis. This procedure resulted in selection of six female and three male colonies. Coral fragments from the female colonies were distributed in two aquaria, referred to as spawning aquaria. Subsequently the fertile male colony with the higher number of available fragments was selected and four of its fragments were distributed in each of the two spawning aquaria. The remaining male colonies were not used to avoid polyspermy (*Levitan, Terhorst & Fogarty, 2007*).

To increase the potential of spawning, we enriched the aquaria water with free mature sperm, obtained from the selected male colony. This was achieved by dissecting mature spermatocysts from coral tissue, which were subsequently concentrated in 50 ml flasks with filtered (mesh size: 0.2 μm) SW, mixed by gently shaking and redistributed to the aquaria. Water inflow was paused and aquaria pumps were substituted with aeration to ensure water circulation without losing or harming potentially spawned gametes. Upon gamete release, which happened in batches separated by intervals of at least 2–3 h, gametes/fertilized eggs from each batch were collected from the water column to a 750 ml-culture flask (20–100 fertilized eggs per flask), filled with filtered SW from the aquaria facilities (mesh size: 0.2 μm). Whenever more than 100 gametes/embryos were released in one batch, these were equally distributed to two flasks to avoid maintaining larvae in high densities. During the first four days of the study we collected a total of 688 gametes which were distributed to 7 batches. Three of these batches were large enough to be split to two flasks (total $n = 10$ flasks).

## Temperature experiments

In order to choose appropriate temperature regimes for larval rearing, we utilized temperature data collected during annual CTD surveys, under the framework of the projects CONDOR (EEA Grants PT-0040) and SMaRT (SRECC- Azores Regional Government M.2.1.2/029/2011). Data were collected between 2010 and 2012, above the coral garden where specimen collection took place. Subsequently, we utilized the minimum and maximum recorded values during the spawning season of the target species (October-November) to define the target rearing temperatures (13 ±0.5 °C and 15 ±0.5 ° C). Two water baths were set-up, each maintaining temperature within ±0.5 °C of the corresponding target temperature, with the aid of an aquaria chiller and a heater, respectively. Each day, the collected batches were divided between the two temperature treatments: immediately after collection of the released fertilized eggs/embryos, culture flasks were randomly assigned to one of the two water baths ($n = 5$ in each water bath). This corresponded to a total of 346 and 342 embryos reared at 13 °C and 15 °C respectively. Culture flasks were equipped with glass pipettes connected to an aquaria air pump, achieving continuous light circulation, while the full volume of water in the flasks was exchanged daily.

## Embryonic and larval development

Embryos were monitored every 3–4 h during the first 48 h and subsequently once a day until reaching the planula stage, to study their early development. In every monitoring event, all embryos were counted to estimate survival. Additionally, 10–15 embryos were randomly removed from each flask and photographed, with a digital camera (DIGICAM 5MEG LCMOS MAC) attached to a microscope (10×), to record their developmental stage and size. Embryos were subsequently returned to the flasks. Due to the sometimes prolonged gamete release, gametes of the same batch were occasionally in slightly different developmental stages, therefore the timing of embryonic development is approximate. Moreover, since it was not possible to define the moment of fertilization, embryo development is presented in respect to the time of gamete release. To estimate size,

we measured width and length (mm) of embryos and larvae (days 4 and 14) using the open software Fiji/Image J (*Schindelin et al., 2012*). The data were subsequently used to estimate volume (mm$^3$) assuming larvae had the shape of a prolate spheroid (*Larsson et al., 2014*). The ratio of length to width (LW ratio) was used as a proxy of sphericity.

## Embryo and larval survival

After reaching the planula stage, larvae were counted every 2–3 days. The last count corresponded to day 34, 36 or 39, depending on the batch. The obtained data were joined to the dataset collected during embryo development to estimate larval survival during the whole experimental period. Survival analysis was performed using the Kaplan–Meier method (*Kaplan & Meier, 1958*), following the rationale of *Graham, Baird & Connolly (2008)*. Since monitoring was done in time intervals and the exact time of death for each larva was not known (interval-censored data), we assumed that time of death was the moment at which each larva was observed for the last time. The remaining larvae at the last monitoring event were considered alive (censored data). As the Kaplan–Meier method does not allow for incorporation of replicate information into the analysis, we performed the analysis by pooling data from all batches together, for each rearing temperature. Subsequently the analysis was repeated separately for each batch, to provide information about the variability among batches (*Graham, Baird & Connolly, 2008*). A log-rank test was performed to compare the survival curves between larvae reared under 13 °C and 15 °C. Survival analysis was performed by using the packages survival (*Therneau & Grambsch, 2000*) and survminer (*Kassambara, Kosinski & Biecek, 2019*) in R 3.5.0 (*R Core Team, 2019*).

## Larval swimming behaviour

Data on swimming speed and behaviour were collected by video recording and analysis. Videos were recorded with a Canon EOS 600D digital camera, equipped with a regular 22–55 mm lens, on day 4 and day 15 after spawning, which corresponded to the first day larvae reached the mature planula stage and the second day larvae started settling, respectively. To minimize larval handling, swimming behaviour was recorded in the same culture flasks used for larval rearing. Videos were captured in the dark, using lateral led lights for illumination (Stromberg & Larsson, 2017). Flasks were positioned in front of a black slide with a calibrated grid that was used as background and a 2-minute waiting period was implemented to ensure no water movement was interfering with larval swimming. Subsequently, three videos (duration: 1 min) were recorded at three minute intervals.

Videos were converted to frames and were analyzed by an automatic particle tracking method, using the open software Fiji/Image J (*Schindelin et al., 2012*) and the plugin TrackMate (*Tinevez et al., 2017*) to record data on vertical swimming behaviour, namely swimming direction (up/down), displacement and swimming speed. Estimates of swimming speed only considered tracks with displacement higher than 2 mm, to exclude data from larvae that did not move or moved minimally.

## Pelagic phase and larval settlement

During the counts performed for survival, each larva was assigned to one of four stages: planula, settled, pelagically metamorphosed and deformed. Because counts were made simultaneously for all flasks and each flask contained a batch of different age, e.g., some batches were released with 1–3 day difference, when average counts were estimated these were sometimes heavily influenced by the available count for that day. To be able to estimate robust mean counts for each monitoring day, missing counts were regenerated for each batch separately by using linear interpolation between existing data points (*Dong & Peng, 2013*), by using the R package VIM (*Kowarik & Templ, 2016*). Extrapolation was performed only until the last datapoint that was available for each batch, i.e., there was no attempt to predict the trend past the last available count. Subsequently, counts of each stage were divided by the total number of living larvae in each batch. This resulted in estimates of the proportion of the surviving larvae in each stage and was used to analyze the behaviour of the remaining larvae. Lastly, on days 4 and 14 after spawning, five planulae were removed from each flask (total $n = 25$ for each temperature regime) and photographed with a digital microscope camera to estimate their size.

Since larvae did not display clear bottom probing behaviour, the onset of competency was defined by settlement or pelagic metamorphosis. After the first larval settlement (day 14), substrate was provided to the culture flasks in order to monitor settlement behaviour. Three flasks from each temperature regime were randomly selected and three pieces (approximate diameter: 5 mm) of basalt rock attached to a plastic slide (10 mm ×80 mm) were offered as potential substrate in each flask. Basalt was selected because it is an abundant hard substrate in the deep seafloor of the Azores and where the studied species is frequently observed. The substrate was not pretreated to develop biofilm. Settled larvae were observed and photographed every 2–3 days to assess and describe settlement and metamorphosis, during a period of approximately two weeks. After metamorphosis was observed, a mixture of live microalgae (*Nannochloropsis gaditana* and *Chaetoceros calcitrans*) and rotifers was provided weekly as a potential food source.

## Statistical analysis

For all the dependent variables in question, we firstly performed exploratory analysis (*Zuur, Ieno & Elphick, 2010*) to select the most appropriate modeling method. The effect of each independent variable was subsequently tested with linear models (LMs), by adding the independent variables progressively to the respective model and using maximum likelihood ratio (MLR) tests and the Akaike Information Criterion (AIC). Data collected from monitoring larvae stages (proportions) were modeled by means of Generalized Additive Models (GAMs) with a binomial distribution. Summarized results of the MLR test for each variable in question are provided in Table 1, while the results from each selected model are provided graphically as supplementary material (Figs. S1–S5). Statistical analysis was performed in R (*R Core Team (2019)*).
**Table 1  Model selection results.** Maximum Likelihood Ratio (MLR) tests reveal significant effects of the independent variables in question. AIC, Akaike Information Criterion; df, degrees of freedom; p, *p* value of the respective anova test among models. Best models are highlighted in grey.

| Dependent variable | Model type | Model | AIC | X$^2$ | df | p |
|---|---|---|---|---|---|---|
| Size | LM | Null | −186.35 | | | |
| | | Stage | −725.87 | 8.63 | 13 | $2.20 \times 10^{-16}$ |
| | | Stage + Temperature | −724.69 | 0.004 | 12 | 0.37 |
| | | Stage × Temperature | −708.56 | 0.03 | 11 | 0.90 |
| Length/width ratio | LM | Null | 184.68 | | | |
| | | Stage | −132.57 | 21.66 | 13 | $2.20 \times 10^{-16}$ |
| | | Stage + Temperature | −130.58 | 0.0004 | 12 | 0.91 |
| | | Stage × Temperature | −159.44 | 1.66 | 11 | $4.78 \times 10^{-7}$ |
| Swimming speed (13 °C) | LM | Null | −141.32 | | | |
| | | Time | −170.05 | 1.24 | 1 | $1.95 \times 10^{-8}$ |
| | | Time + Direction | −168.18 | 0.005 | 1 | 0.71 |
| | | Time × Direction | −166.18 | 0.00001 | 1 | 0.99 |
| Swimming speed (15 °C) | LM | Null | 77.12 | | | |
| | | Time | 63.7 | 1.02 | 1 | $7.91 \times 10^{-5}$ |
| | | Time + Direction | 64.11 | 0.10 | 1 | 0.20 |
| | | Time × Direction | 65.75 | 0.23 | 1 | 0.54 |
| Swimming speed | LM | Null | 42.04 | | | |
| | | Time | −24.12 | 4.00 | 1 | $2.20 \times 10^{-16}$ |
| | | Time + Temperature | −79.80 | 3.17 | 1 | $1.41 \times 10^{-14}$ |
| | | Time × Temperature | −77.89 | 0.04 | 1 | 0.767 |
| Swimming direction (13 °C) | LM | Null | 80.31 | | | |
| | | Time | 82.31 | 0 | 1 | 1 |
| | | Time + Direction | 84.15 | 5.35 | 1 | 0.70 |
| | | Time × Direction | 82.90 | 95.15 | 1 | 0.11 |
| Swimming direction (15 °C) | LM | Null | 89.65 | | | |
| | | Time | 91.65 | 0.00 | 1 | 1 |
| | | Time + Direction | 92.38 | 88.60 | 1 | 0.34 |
| | | Time × Direction | 94.38 | 0.09 | 1 | 0.97 |
| Proportion of planula | Binomial GAM | Null | 4260.04 | | | |
| | | Time | 811.09 | 3454.09 | 2.57 | $2.20 \times 10^{-16}$ |
| | | Time + Temperature | 749.43 | 63.61 | 0.97 | $1.5 \times 10{-}15$ |
| | | Time × Temperature | 749.68 | 2.06 | 1.16 | 0.15 |
| Proportion of metamorphosed | Binomial GAM | Null | 2658.9 | | | |
| | | s(Time, $k = 4$) | 467.12 | 2196.8 | 2.52 | $2.20 \times 10^{-16}$ |
| | | s(Time, $k = 4$) + Temperature | 468.32 | 0.80 | 1 | 0.36 |
| | | s(Time, $k = 4$, by=Temperature) | 465.44 | 6.68 | 1.9 | 0.03 |
| Proportion of settled | Binomial GAM | Null | 1359.07 | | | |
| | | s(Time, $k = 4$) | 670.27 | 694.69 | 2.95 | $2.20 \times 10^{-16}$ |
| | | s(Time, $k = 4$) + Temperature | 578.11 | 94.11 | 0.97 | $2.20 \times 10^{-16}$ |
| | | s(Time, $k = 4$, by=Temperature) | 564.18 | 17.74 | 1.9 | $1.40 \times 10^{-4}$ |

**Table 1** (*continued*)

| Dependent variable | Model type | Model | AIC | X² | df | p |
|---|---|---|---|---|---|---|
| Proportion of deformed | Binomial GAM | Null | 551.37 | | | |
| | | s(Time, $k = 4$) | 220.45 | 335 | 2.07 | $2.20 \times 10^{-16}$ |
| | | s(Time, $k = 4$) + Temperature | 220.05 | 2.44 | 1.02 | 0.11 |
| | | s(Time, $k = 4$, by=Temperature) | 184.87 | 36.86 | 0.84 | $1.26 \times 10^{-9}$ |

## RESULTS

### Spawning

Gamete release occurred for the first time on the 27th of November, one day after the new moon. Oocytes were encountered 15 min after enrichment with free live sperm, in both aquaria. Spawning was not synchronized among colonies, neither among polyps of the same colony. Despite careful observation, it was not possible to directly observe polyps releasing sperm or oocytes and determine whether one or more colonies participated in gamete release. Similarly, it was not possible to directly observe if fertilization was internal or external. All collected oocytes were fertilized, therefore fertilization was either internal, or external with very high fertilization rates. Oocytes were spherical, they had no visible germinal vesicle and were released in batches of 10-80 at a time. They were mostly negatively buoyant, however, they remained in suspension for several hours due to water movement within the aquaria. Average oocyte diameter was 365.4 $\pm$24.2 $\mu$m. Gamete release was slow and sometimes continued for 1–3 h. It happened multiple times a day (every 2–3 h) for a week and continued with lower frequency (every 1–3 days) for approximately a month. Release occurred both during day and night hours and did not seem to follow any circadian pattern.

### Embryonic and larval development

Cell division was always equal but cleavage varied highly among stages and embryos. It was not possible to determine the timing of the first division after spawning. Cytokinesis was never visible for the 2-cell stage, in which cleavage seemed to be always superficial (Fig.1B). During the following stages, cleavage varied from radial to pseudospiral and in some cases superficial, leading to embryos with substantial differences in shape. Development always led to a hollow blastula (Fig. 1G) followed by gastrulation and the formation of planula larvae without visible oral pore (Fig. 1I). Cleavage and cell division did not differ between the two rearing temperatures.

At 13 °C, all embryos reached the blastula stage within 10 h and the early gastrula stage within 48 h (Fig. 2). After 72 h all embryos reached the late gastrula stage and could perform slow, mainly rotating movements by cilia, while fully competent, swimming planulae were formed after 96 h (4 days). During their development, embryos were negatively buoyant and accumulated at the bottom of the flasks. In the first batch this resulted in the formation of embryo aggregations and abnormal embryo development. This issue was solved by adding slight aeration that ensured water and oxygen circulation within the flasks. At 15 °C, during the first 6 h cleavage seemed to be occurring at similar intervals until reaching

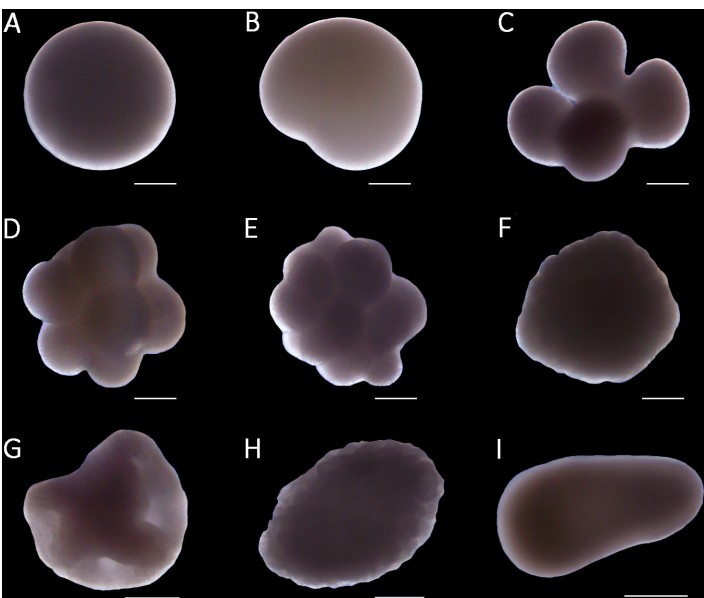

**Figure 1 Stages of embryo development of the octocoral species *Dentomuricea* aff. *meteor*.** (A) fertilized oocyte. (B) 2-cell. (C) 4-cell. (D) 8-cell. (E) 16-cell. (F) 64-cell. (G) hollow blastula. (H) gastrula. (I) planula.

the blastula stage (Fig. 2), however, embryos reached the late gastrula and subsequently the planula stage approximately 24 h (after 72 h) earlier than embryos reared at 13 °C (Fig. 2).

Embryos between the 2-cell and 32-cell stage obtained variable shapes (Fig. 1) and their volume was on average $0.03 \pm 0.0073$ mm$^3$. Subsequently, during the 64-cell stage and blastula they turned more spherical but had a similar volume range ($0.03 \pm 0.005$ mm$^3$). After reaching the planula stage, embryos increased significantly in size (Table 1) and planulae reached $0.28 \pm 0.1$ mm$^3$ on day 4 and $0.67 \pm 0.28$ mm$^3$ on day 14, with measurements on day 14 displaying substantial variability. Mature planulae displayed the capacity to change their shape between spherical and elongated, and more elongated larvae were observed on day 14 compared to day 4 (Fig. S1). This was also confirmed from the LW ratio which presented a non-significant decrease from late gastrula embryos ($1.49 \pm 0.17$ mm$^3$) to planulae on day 4 ($1.22 \pm 0.29$ mm$^3$) but increased significantly (Table 1) on day 14 ($2.02 \pm 0.45$ mm$^3$). Embryo sizes were not statistically different between the two temperatures (Table 1). Planulae on day 4 had significantly higher LW ratios at 15 °C (LW = $1.59 \pm 0.39$; Table 1), showing a tendency to maintain a more elongated shape than at 13 °C (Fig. S1).

## Embryo and larval survival

In both temperatures, survival differed substantially among batches (Fig. S2). In most batches reared at 13 °C, a sharp decline in survival rates was observed during the first 48 h, after which a more moderate mortality rate was established (Fig. 3). In the same temperature treatment, median survival time, i.e., time when mortality reached 50%, was 11 days while survival after 36 days was 16.4%. At 15 °C, the average mortality rate

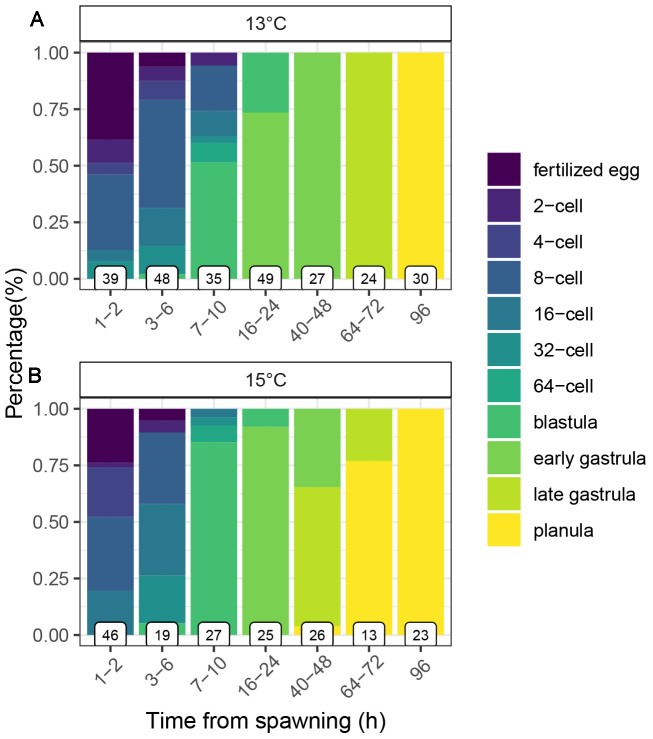

**Figure 2** **Early development of embryos of the octocoral species *Dentomuricea* aff. *meteor* reared under 13 °C (A) and 15 °C (B).** Bars display the proportion of embryos in each developmental stage over a course of 96 h after spawning. Numbers at the base of each bar represent the sample size (n).

seemed to be more constant (Fig. 3). Median survival time was 5 days longer than at 13 °C (16 days), however, final survival after 36 days was slightly lower (12.6%). Overall, these differences were not statistically significant according to the log-rank test ($p = 0.05$; Fig. 3).

## Swimming behaviour

Planulae remained mostly at the bottom of the culture flasks, where they displayed slight rotational and unidirectional movements. They rarely became waterborne without the aid of water movement. Once in the water column, larvae did not show a specific swimming pattern but followed random trajectories. Overall, for larvae reared under 13 °C, 51.2 ±14.2% of the recorded larval tracks were directed upwards while 50.7 ±6.33% were directed downwards. It was not clear if downward movement involved swimming or just sinking. The proportion of upward/downward swimming larvae did not change significantly with time (Table 1). Larvae displayed an average swimming speed of 0.24 ±0.16 mm s$^{-1}$ on day 4 and 0.36 ±0.21 mm s$^{-1}$ on day 15. Swimming speed did not differ significantly between upward and downward movements (Table 1) but it was significantly higher on day 15 compared to day 4 (Table 1).

Swimming velocity for larvae reared under 15 °C was similar between upward and downward swimming (Table 1) and increased slightly but significantly with time (Table

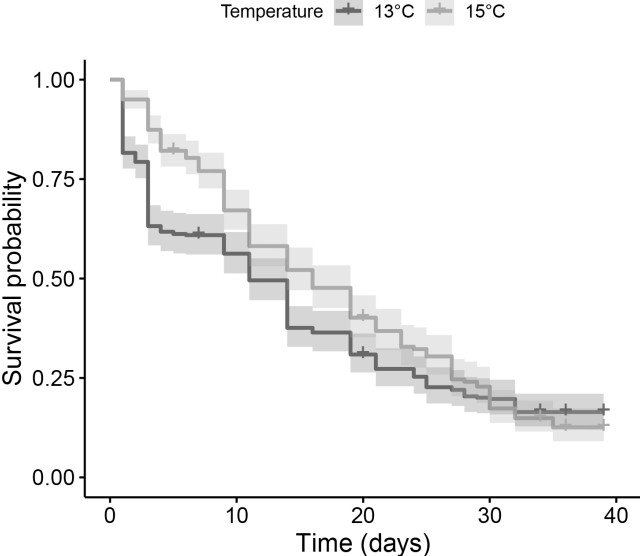

**Figure 3  Comparison of Kaplan–Meier estimates of larvae survival of the species *Dentomuricea* aff. *meteor* under two temperature regimes.** The initial pool of embryos corresponded to 346 and 342 embryos at 13 °C and 15 °C, respectively.

1), from 0.4 ±0.24 mm s$^{-1}$ on day 4 to 0.44 ±0.23 mm s$^{-1}$ on day 15. Overall, 52.7% of the recorded tracks were directed downwards and the proportion of upward/downward swimming tracks did not differ significantly between dates (Table 1). Larvae swimming velocity was significantly higher under 15 °C compared to 13 °C, (Fig. 4) both on day 4 and day 15 (Table 1).

## Pelagic phase and settlement

The proportion of planulae decreased substantially during the course of the experiment, mainly due to high mortality (Fig. 4A). The surviving planulae followed slightly different trends between the two temperatures with planulae under 15 °C remaining in the pelagic phase for a longer period (Fig. 4B), a difference that was statistically significant (Table 1). In both temperatures, after day 36 only a minimal proportion of larvae remained (Fig. 4A) and the last free swimming planulae were observed on day 39.

Larvae started settling on day 14 under 13 °C and on day 17 under 15 °C. Under both experimental temperatures, larvae settled on the flask walls and plastic slides whereas no larvae attached to the provided basalt rock. Since the addition of substrate did not have any effect on settlement behaviour, data from all flasks, i.e., with and without provided substrate, were pooled together for further analysis. All settled larvae underwent metamorphosis. Larvae firstly obtained a pear-like shape and subsequently became rounder, gradually forming a polyp base, mouth and mesenteries (Fig. 5A). Fully developed primary polyps were formed within approximately 2–3 days, after the formation of tentacles, sclerites and tentacle pinnules (Fig. 5B). In both rearing temperatures, the number of settled larvae corresponded to a very low proportion of the initial pool of planulae, corresponding to 3.21% (11 larvae) under 13 °C and 1.46% (5 larvae) under 15 °C. Nevertheless, surviving

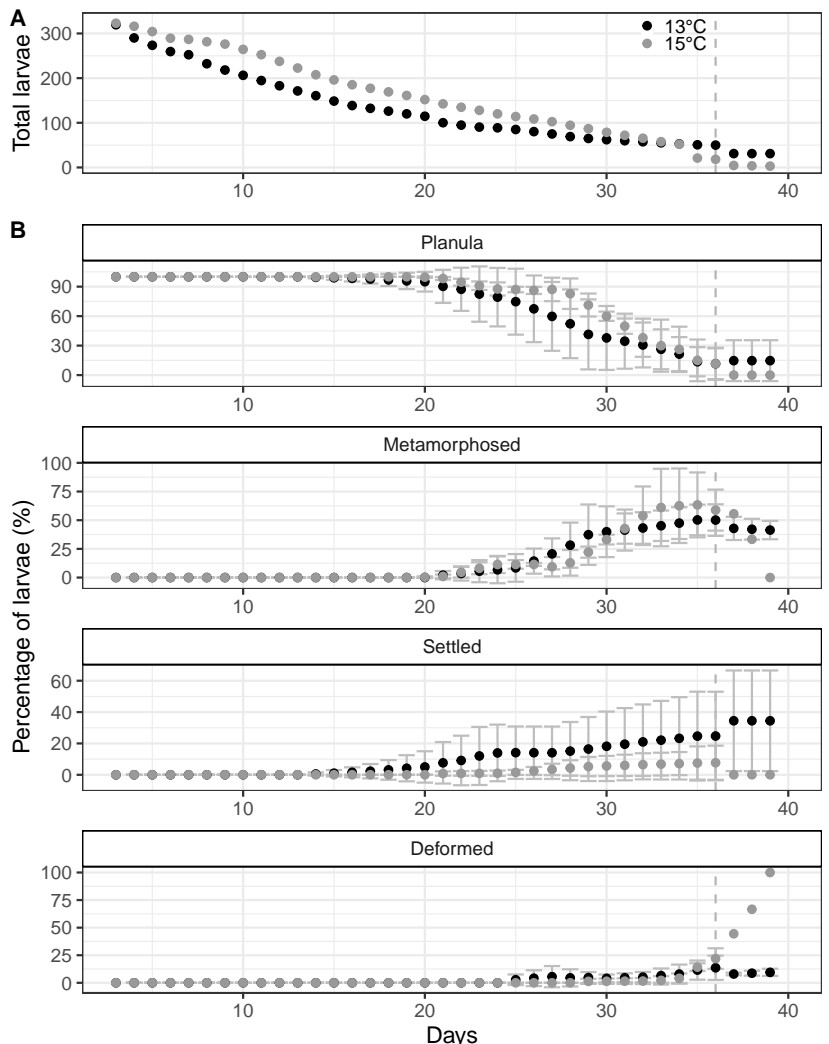

**Figure 4 Larval behaviour of the octocoral species *Dentomuricea* aff. *meteor* during the pelagic phase under two experimental rearing temperatures.** (A) Total number of surviving larvae in each rearing temperature. (B) Proportion of larvae in different developmental stages (planula, metamorphosed but not settled, settled, deformed) under two experimental rearing temperatures. Dotted vertical line represents the last timepoint when data for all batches were available.

planulae displayed slightly but significantly different trends during the course of the study (Table 1), with a larger proportion of larvae settling earlier under 13 °C than under 15 °C (Fig. 4). A high variance was observed on the estimates of the average proportion of settled larvae (Fig. 4) among batches at 13 °C, mainly due to a single batch in which very few larvae settled throughout the study period.

After day 20, an increasing proportion of the surviving larvae initiated metamorphosis without settling (Fig. 4), in both temperature regimes. This form of pelagic metamorphosis started with planula larvae obtaining a pear shape (Fig. 6A) and continued with formation of mouth, mesenteries, tentacles and finally sclerites (Figs. 6B, 6C). Metamorphosis from

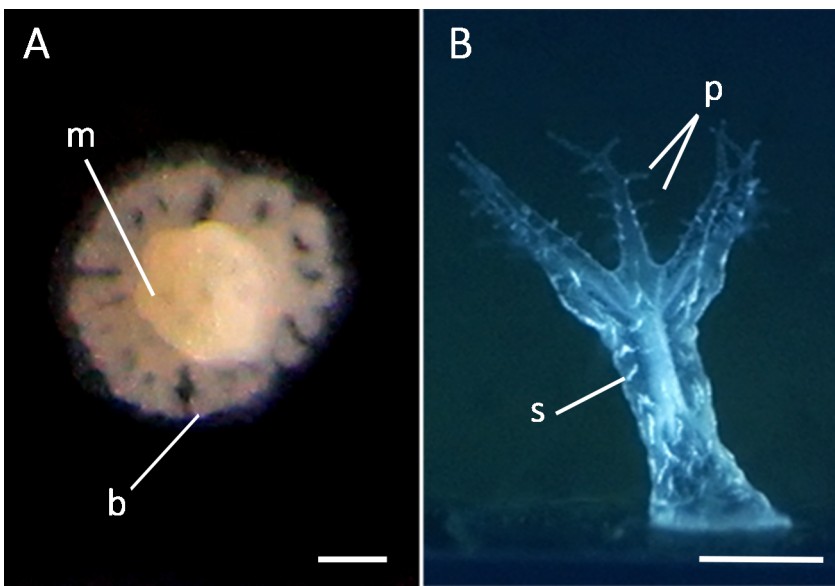

**Figure 5** Formation of primary polyps from settled planulae of the octocoral *Dentomuricea* aff. *meteor.* (A) Recently settled primary polyp with a polyp base (*b*) and formation of eight mesenteries (*m*). (B) Final primary polyp with sclerites (*s*), tentacles and tentacle pinnules (*p*). Scale bar: 500 µm.

planula larva to primary polyp took approximately 2–3 days. None of the larvae that displayed pelagic metamorphosis settled during the course of the study. Metamorphosed larvae were still able to get transported by water movements but displayed limited swimming ability. The trend of pelagically metamorphosed larvae appeared to be significantly different between the two temperatures (Table 1), but the constructed model was heavily influenced by one batch under 15 °C in which all remaining planulae on day 31 metamorphosed pelagically and subsequently presented deformations and deceased (Fig. 4B). Overall, during the experimental period 26 larvae metamorphosed pelagically under 13 °C and 28 under 15 °C, representing only 7.5% and 8.18% of the initial planulae pool. Deformed larvae were observed in both temperatures but represented a small proportion of the initial pool (2.02% under 13 °C and 1.16% under 15 °C). Under 13 °C, they started appearing on day 24 (Fig. 4B) but remained in low numbers throughout the experimental period. Under 15 °C, they appeared 2 days later, but reached significantly higher proportions after day 35 (Table 1, Fig. 4B). Most of these late deformations under 15 °C were observed in pelagically metamorphosed larvae (Fig. 6D).

## DISCUSSION

So far, studies on the biology and ecology of deep-sea octocorals have focused mainly on the adult stage (*Watling et al., 2011*), with very few studies tackling early life history stages (*Cordes, Nybakken & Van Dykhuizen, 2001*; *Sun, Hamel & Mercier, 2010*; *Sun, Hamel & Mercier, 2011*). To our knowledge, the present study is the first to provide a detailed insight to the larval biology of a deep-sea octocoral species including embryo and larval

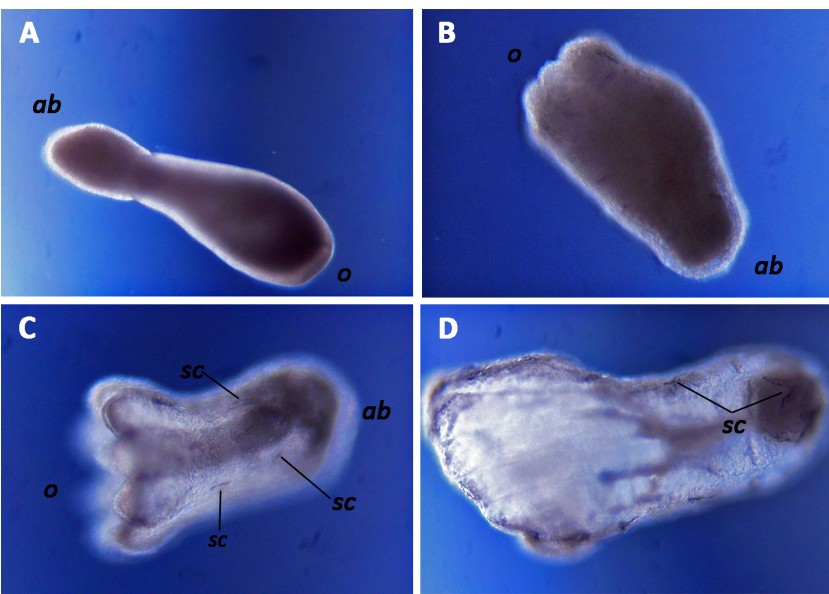

**Figure 6** **Pelagic metamorphosis of planulae of the octocoral species *Dentomuricea* aff. *meteor*.** (A) pear shaped larva with formed mouth at the oral side (*o*) and closed aboral side (*ab*). (B) Tentacle formation on the oral side. (C) Fully formed tentacles, mesenteries and sclerites (*sc*). (D) Deformed larva with abnormal mesentery and tentacle formation.

development, larval survival, swimming and settlement behaviour, which are essential variables to understand dispersal and connectivity in the deep-sea (*Gary et al., 2020*).

In our study, it was not clear if spawning was actually induced, assisted or just coincided with sperm enrichment, due to the limited time interval between sperm enrichment and the first gamete release. Repetitive release of gametes or planulae within a specific period is common among octocorals, including tropical broadcast spawning (*Pakes & Woollacott, 2008*; *Wells, Tonra & Lasker, 2020*), temperate brooding (*Weinberg & Weinberg, 1979*; *Martínez-Quintana et al., 2014*) and deep-sea brooding species (e.g., *Sun, Hamel & Mercier, 2011*). This strategy may increase the probability that some embryos and larvae develop under optimal conditions (*Kahng et al., 2008*). In species with this behaviour, studying the effects of different environmental variables on embryo development is crucial, as embryos of different cohorts are likely to be released under different environmental conditions including temperature, salinity, pH and food availability.

Embryo and larval development of *Dentomuricea* aff. *meteor* had many similar characteristics with other octocoral species. Unequal cleavage that ranges from radial to pseudospiral is common among cnidarians (*Fritzenwanker et al., 2007*), including tropical brooding (*Benayahu & Loya, 1983*; *Dahan & Benayahu, 1998*) and broadcast spawning (*Mandelberg-Aharon & Benayahu, 2015*) octocorals. Superficial cleavage, like the one observed in *D.* aff. *meteor* is frequently encountered in embryos with high amounts of yolk reserves (*Scriba, 2015*), indicating a lecithotrophic larvae which was also confirmed by the absence of oral opening before metamorphosis. Larval size was also comparable to that of other octocoral species (Table 2). Overall, these findings suggest that some reproductive

and larval characteristics might be conserved among taxonomically related groups, despite local adaptations due to depth and other habitat limitations.

Temperature is considered one of the main factors affecting larval biology, with higher temperatures usually resulting in faster developmental rates (*Hoegh-Guldberg & Pearse, 1995*). Our results were consistent with this premise, with larvae reaching the planula stage 24 h earlier at 15 °C when compared to 13 °C. This difference in developmental time is likely the drive between the differences in the LW ratios, since planulae tend to be more elongated with age. The different developmental rates did not affect survival which was similar and very low for both temperatures but varied substantially among batches. Since it was not possible to observe which female colonies participated in each gamete release, the possibility that first batches had lower survival cannot be excluded. Larval characteristics such as size and longevity have been shown to vary between cohorts in many marine larvae (*Marshall, Bonduriansky & Bussière, 2008*; *Cumbo, Fan & Edmunds, 2012*; *Martínez-Quintana et al., 2014*). Such variability between offspring has been considered an adaptive strategy to increase offspring survival in species that inhabit unstable environments (*Cooper & Kaplan, 1982*; *Marshall, Bonduriansky & Bussière, 2008*).

Most deep-sea octocoral planulae studied so far, displayed low mobility, negative buoyancy and crawling (e.g., *Drifa glomerata*, *Sun, Hamel & Mercier, 2010*; *Duva florida*, *Sun, Hamel & Mercier, 2011*) or very limited swimming capacity (e.g., *Drifa sp.*, *Sun, Hamel & Mercier, 2010*). On the contrary, larvae of *D.* aff. *meteor* were active swimmers but initiated swimming only after stimulation, a behaviour also recorded in *Corallium rubrum* (*Martínez-Quintana et al., 2014*). Swimming in *D.* aff. *meteor* was random, as revealed by the similar proportion of upward and downward swimming. When compared to other deep-sea broadcast spawning corals, such as the scleractinian *Lophelia pertusa*, *D.* aff. *meteor* had lower swimming capabilities, especially since *L. pertusa* displayed intense, negative geotactic behaviour (*Larsson et al., 2014*). Swimming velocity of *D.* aff. *meteor* was comparable to that of *C. rubrum* (Table 2) and *L. pertusa* (*Larsson et al., 2014*), but larvae of these species were maintained under different temperature regimes (19–20 °C for *C. rubrum*, (*Martínez-Quintana et al., 2014*; 8–12 °C for *L. pertusa*, *Larsson et al., 2014*). Temperature can affect both larval physiology and water characteristics since higher temperature often causes a decrease in viscosity and increase in larval metabolic rates (*Von Herbing, 2002*). Both effects can result in higher swimming velocity and are likely to be associated with the higher larval swimming speed of *D.* aff. *meteor* under 15 °C. Nonetheless, metabolism is not the only physiological process affected by temperature and larvae display physiological limits, which need to be further studied for the target species.

Larval planktonic period can be divided in two phases, an obligatory phase that lasts until the onset of developmental competence (the ability to respond to settlement cues) and a facultative phase that depends on settlement behaviour in response to the existence of certain substrate characteristics (competency window, *Elkin & Marshall, 2007*). In the present study, both phases were characterized by high mortality, leading to a loss of more than 50% of planulae before the defined onset of competency. Moreover, the onset of competency was inferred by the first larval settlement since larvae did not display any specific geotactic or bottom probing behaviour, but it is possible that larvae had entered

**Table 2  Summary of embryo and larval characteristics of octocoral species in the order Alcyonacea.** Depth: deep (>200 m) and shallow (<200 m). Reproductive mode: internal brooding (IB), broadcast spawning (BS), surface brooding (SB). T: temperature at which larvae were reared. Larval size is presented as length (mm). Competency refers to the period when larvae are competent to settle. Variables are provided either as range, average ±standard deviation, maximum (max) or median (median) values.

| Family | Habitat | Depth | Repr. mode | Species | T (°C) | Larval size (mm) | Competency (days) | Longevity (days) | Swimming behaviour | Swimmingspeed (cm/s) | Reference |
|---|---|---|---|---|---|---|---|---|---|---|---|
| Alcyoniidae | Temperate | Deep | IB | *Anthomastus ritteri* | | 3.3 ±1 | 2–3, 123$^{max}$ | | | | *Cordes, Nybakken & Van Dykhuizen (2001)* |
| Coralliidae | Temperate | Shallow | IB | *Corallium rubrum* | 22 | 1.5 | | | | | *Weinberg & Weinberg (1979)* |
| | | | | | 19–21 | 1.0 | | 28.9 ±3.3 | Vertical swimming | 0.045–0.056 | *(Martínez-Quintana et al., 2014)* |
| Gorgoniidae | Tropical | Shallow | BS | *Antillogorgia americana* | 24 | | 36$^{median}$ | >60 | Vertical swimming | 0.22 ±0.01$^{max}$ | *Coelho & Lasker (2016)* |
| | Temperate | Shallow | IB | *Eunicella singularis* | 22 | 2.5 | | | | | *Weinberg & Weinberg (1979)* |
| | | | | | 18–20 | | | 35.0 ±11.6 | | | *Guizien et al., (2020)* |
| | Tropical | Shallow | IB | *Parerythropodium f. Fulvum* | 21–26 | | 1–64 | 76$^{max}$ | | | *Ben-David-Zaslow & Benayahu, (1998)* |
| | Tropical | Shallow | BS | *Dendronephthya hemprichi* | 21–26 | | 2–74 | 81$^{max}$ | | | *Ben-David-Zaslow & Benayahu (1998)* |
| | Tropical | Shallow | IB | *Litophyton arboreum* | 21–26 | | 1–57 | 92$^{max}$ | | | *Ben-David-Zaslow & Benayahu (1998)* |
| | Tropical | Shallow | IB | *Nephthea sp.* | 21–26 | | 1–57 | | | | *Ben-David-Zaslow & Benayahu (1998)* |
| Nephtheidae | Subarctic | Deep | IB | *Gersemia fruticosa* | | 1.5–2.5 | 40–70 | | Swimming | | *Sun, Hamel & Mercier (2011)* |
| | Subarctic | Deep | IB | *Duva florida* | 0-9 | 1.0–2.5 | 5 | | Crawling | | *Sun, Hamel & Mercier (2011)* |
| | Subarctic | Deep | IB | *Drifa glomerata* | 2 | 4.0–5.0 | | | | | *Sun, Hamel & Mercier (2010)* |
| | Tropical | Shallow | BS | *Plexaura kuna* | 28–30 | 2.0 | 4–21 | | | | *Lasker & Kim (1996)* |
| | Temperate | Shallow | SB | *Paramuricea clavata* | 18–20 | | | 32 ±11 | Crawling | | *Guizien et al., (2020)* |
| Plexauridae | Tropical | Shallow | BS | *Plexaura homomalla* | 27–29 | 1.0 | 4 | | Swimming and crawling | 0.5 | *Wells, Tonra & Lasker (2020)* |
| | Temperate | Deep | BS | *Dentomuricea aff. meteor* | 13 | 1.15 ±0.28 | 25 | 11 | Swimming and crawling | 0.024–0.036 | This study |
| | | | | | 15 | 1.14 ±0.28 | 29 | 16 | Swimming and crawling | 0.04–0.044 | This study |
| Xeniidae | Tropical | Shallow | IB | *Xenia umbellata* | 21–26 | | 2–76 | 155$^{max}$ | | | *Ben-David-Zaslow & Benayahu (1998)* |
| | Tropical | Shallow | IB | *Heteroxenia tuscescens* | 21–26 | | 49$^{max}$ | 50$^{max}$ | | | *Ben-David-Zaslow & Benayahu (1998)* |

competency before actually settling. Settlement rates were low and a higher proportion of the surviving larvae metamorphosed without settling. These are strong indications that adequate settlement surfaces and cues were not provided during the study. It is thus likely that larvae were forced to proceed to the next ontogenetic phases (settlement and metamorphosis) due to the lack of energy reserves. This phenomenon has been tentatively explained by the "desperate larvae hypothesis" (*Gibson, 1995*; *Marshall & Keough, 2003*), which states that the duration of the planktonic phase is likely determined by the availability of energetic reserves (*Wendt, 2000*) and therefore non-feeding larvae can only delay settlement and metamorphosis until reaching a specific reserve level (*Elkin & Marshall, 2007*).

Remarkably, settlement only took place on plastic surfaces while none of the larvae attached on the provided basalt rock. This was slightly unexpected since *D.* aff. *meteor* has been observed to colonize basalt rock in seamounts in the Azores. It is highly possible that this was due to the lack of bacterial biofilm on the rock, which has been shown as an important settlement clue for other invertebrates (*Hadfield, 2011*). Moreover, the provided rock occupied a very small area compared to the flask walls. Settling on plastic is not uncommon among octocorals (*Lasker & Kim, 1996*; *Freire et al., 2019*; *Carugati et al., 2021*) but further studies with more settlement surfaces are essential to clarify the settlement requirements of the target species. Pelagic metamorphosis of planulae into polyps has also been reported for many octocorals from shallow tropical (*Ben-David-Zaslow & Benayahu, 1998*; *Lasker & Kim, 1996*), to temperate (*Linares et al., 2008*) and deep-sea species (*Sun, Hamel & Mercier, 2011*). In some corals, pelagic polyps can display high survival and dispersal potential (*Mizrahi, Navarrete & Flores, 2014*) and have the ability to feed (*Ben-David-Zaslow & Benayahu, 1998*; *Linares et al., 2008*). In our study, pelagic polyps displayed high mortality but this could be due to the absence of sufficient or adequate food sources. Nevertheless, pelagic metamorphosis might provide a way to acquire feeding structures and allows the acquisition of energy while waiting for the right settlement cue. In the case of *D.* aff. *meteor*, the high proportion of surviving larvae that displayed this behaviour supports the hypothesis that larvae had limited energy reserves and possibly reached their maximum longevity during the experiment.

Under higher temperature, larvae of *D.* aff. *meteor* remained longer in the pelagic phase and displayed lower settlement rates. This was contrary to the expected outcome, since the higher developmental rates observed under higher temperature are expected to be accompanied by earlier competency and higher settlement rates (*O'Connor et al., 2007*; *Heyward & Negri, 2010*). Faster developmental rates, accompanied by decreased settlement under higher temperatures (+ 3 °C) has been also reported for the tropical octocoral *Heliopora coerulea* (*Conaco & Cabaitan, 2020*). It is possible that these results are related to temperature-induced changes in developmental and physiological mechanisms that were not evaluated in our study. For example, it is possible that faster development under higher temperature was accompanied by faster metabolic rates (*O'Connor et al., 2007*) and resulted in faster consumption of reserves, leading to high rates of pelagic metamorphosis and deformations under the absence of proper settlement cues. Ontogeny depends on certain developmental processes and their timing and while developmental rate can be

plastic, changes in timing are likely to have consequences on structure and function, ultimately affecting individual performance (*Kováč, 2002*).

Overall, the embryonic and larval characteristics of *D.* aff. *meteor* suggest a higher dispersal potential than most deep-sea octocorals studied so far (Table 2). However, when compared to other deep-sea species, the dispersal capacity of *D.* aff. *meteor* appears to be limited. For example, the scleractinian *L. pertusa* delayed the onset of competency up to 3–5 weeks from spawning, displayed active upward swimming and survived without settlement for approximately a year (*Larsson et al., 2014*). Similarly, other deep-sea species such as the bivalve *Bathymodiolus childressi* and the gastropod *Bathynerita naticoidea* display longer longevities (approximately one year) and enhanced upward swimming which indicate much higher dispersal potential than *D.* aff. *meteor* (*Arellano & Young, 2009*). The larvae of these deep-sea species are planktotrophic and therefore are not constrained by reserve availability. Our results highlight that the energy reserves of *D. aff. meteor* are a great limitation for many of its larval traits, especially its longevity and behaviour regarding settlement and metamorphosis. While its swimming behaviour is very likely to allow it to disperse among regional seamounts with the aid of local hydrodynamics, its short longevity is indicative of its narrow regional distribution in the North Mid-Atlantic Ridge, especially when compared with the wide distributions of *L. pertusa* and *B. childressi*.

Since the two temperature regimes used in this study are likely to be experienced by embryos of the target species in their natural environment, our results highlight how small changes in temperature can affect embryo development and larval characteristics, such as swimming velocity and settlement behaviour. Climate change is expected to cause changes in ocean circulation (*Sweetmann et al., 2017*) which can modify the water mass dynamics and alter the physicochemical characteristics encountered by embryos and larvae (*Przeslawski, Byrne & Mellin, 2015*; *Van Gennip et al., 2017*; *Claret et al., 2018*). Under these circumstances, baseline information on the responses of early life history stages under variable conditions is essential to predict potential effects on dispersal and connectivity. For example, embryos and larvae of the Antarctic echinoderm *Sterechinus neumayeri* can withstand high pressures only under a narrow temperature interval which can be encountered in specific water masses that allowed the species to disperse to greater depths (*Tyler, Young & Clarke, 2000*). In the case of this species, potential changes in regional circulation, may affect or even disrupt connectivity between shallow and deeper populations. Moreover, larval dispersal and success are important features not only from an ecological but also from an evolutionary perspective, as their adaptive significance can define the selection of reproductive strategies such as reproductive timing (*Crowder et al., 2014*; *Fan et al., 2017*). In deep-sea corals, reproductive timing has been discussed in relation to the seasonal constraints of adult reproductive physiology (e.g., *Orejas et al., 2002*; *Waller et al., 2014*) but its relation to larval survival and success has not been addressed so far. Further studies on the effect of temperature on larval development, physiology and behaviour are therefore essential to obtain a holistic view of the potential impacts of climate change on deep-sea corals and communities.

## CONCLUSIONS

In our study, we provided a detailed description of embryo and larval characteristics of the species *D.* aff. *meteor*. To our knowledge, this is the first systematic description of the early life history traits of a deep-sea octocoral. Our results suggest that *D.* aff. *meteor* larvae are lecithotrophic with development similar to other octocorals and low dispersal capacity compared to other deep-sea species. Rearing at different temperatures did not affect survival, but significant effects were detected on the rate of embryo development, swimming speed and settlement behaviour which in the field can potentially alter larval dispersal and ultimately success. Deep-sea octocorals are receiving increasing attention as a growing number of studies focus on the habitat requirements and environmental conditions shaping deep-sea communities (*Radice et al., 2016*; *Barbosa, Davies & Sumida, 2020*; *Morato et al., 2020*). However, understanding species distributions requires further knowledge on their early life history biology and dispersal, as these play a key role in the successful occupation of available suitable habitat (*Schurr et al., 2007*; *Robinson et al., 2011*). As attempts of biophysical dispersal modelling are increasing in the deep-sea (*Hilário et al., 2015*; *Ross, Nimmo-Smith & Howell, 2016*), further biological data to feed into these models are essential to obtain a better understanding of deep-sea ecosystems.

## ACKNOWLEDGEMENTS

We are very grateful to Cristina Gutiérrez-Zárate and the scientists and crew of R/V Arquipelago for specimen and data collection, as well as to Eva Giacomello, Gonçalo Graça and Robert Priester for compiling the database of the environmental data used in this study. We received funds and support from the Fundação para a Ciência e a Tecnologia (FCT) through the strategic project (UID/05634/2020) granted to OKEANOS. Maria Rakka is funded by a DRCT PhD grand grant (reference M3.1.a/F/047/2015). António Godinho was supported by the European Union's Horizon 2020 Research and Innovation Programme under MERCES project (grant agreement no. 689518), and Marina Carreiro-Silva was supported by Program Stimulus of Scientific Employment (CCCIND/03346/2020) from the Fundação para a Ciência e Tecnologia. There was no additional external funding received for this study. The funders had no role in study design, data collection and analysis, decision to publish, or preparation of the manuscript.

### Funding

This study was supported by the European Union's Horizon 2020 Research and Innovation Program, under the ATLAS (grant agreement No 678760) and iATLANTIC (grant agreement No 818123) projects.

### Grant Disclosures

The following grant information was disclosed by the authors:
ATLAS: 678760.

iATLANTIC: 818123.

Fundação para a Ciência e a Tecnologia (FCT): UID/05634/2020.

DRCT PhD Grand Grant: M3.1.a/F/047/2015.

European Union's Horizon 2020 Research and Innovation Programme: 689518.

Fundação para a Ciência e Tecnologia: CCCIND/03346/2020.

## Competing Interests

The authors declare there are no competing interests.

## Author Contributions

- Maria Rakka conceived and designed the experiments, performed the experiments, analyzed the data, prepared figures and/or tables, authored or reviewed drafts of the paper, and approved the final draft.
- António Godinho performed the experiments, authored or reviewed drafts of the paper, and approved the final draft.
- Covadonga Orejas conceived and designed the experiments, prepared figures and/or tables, authored or reviewed drafts of the paper, and approved the final draft.
- Marina Carreiro-Silva conceived and designed the experiments, performed the experiments, prepared figures and/or tables, authored or reviewed drafts of the paper, and approved the final draft.

## Data Availability

The code is available in the Supplemental Files. The data is available in the Supplemental Files and at Zenodo: Maria Rakka, Antonio Godinho, Covadonga Orejas, & Marina Carreiro-Silva. (2021). Embryo and larval biology of the deep-sea octocoral Dentomuricea aff. meteor [Data set]. Zenodo. http://doi.org/10.5281/zenodo.5093023.

## Supplemental Information

Supplemental information for this article can be found online at http://dx.doi.org/10.7717/peerj.11604#supplemental-information.

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
