# Peer review of "Embryo and larval biology of the deep-sea octocoral Dentomuricea aff. meteor under different temperature regimes"

_PeerJ, doi:10.7717/peerj.11604_

## Round 0.1 · original submission · Major Revisions

Three expert reviewers have evaluated your manuscript and their comments can be seen below and in attached PDFs. The information and data in the manuscript are valuable however a number of clarifications and corrections need to be made to the manuscript. The discussion section could also be more concise. Please ensure that you attend all comments and suggestions in a revision of the manuscript.

·

Basic reporting

The manuscript fulfils the criteria for basic reporting, with only minor adjustments needed.

Experimental design

The experiment is well defined and performed. I have only one concern, and that is that only one male was used for the reproduction producing the larvae for the experiment. It worked out fine despite this, but it is a bit risky since not all couples produce viable offspring. This is just a note to the authors that they might want to consider using several individuals from each sex for future experiments.

Validity of the findings

No comment, all good.

Additional comments

Congratulations to a successful rearing of deep-sea coral larvae and subsequent experiments. The results are much needed!

·

Basic reporting

Please, see the attached document.

Experimental design

Please, see the attached document.

Validity of the findings

Please, see the attached document.

Additional comments

Please, see the attached document.

Reviewer 3 ·

Basic reporting

See comments in General comments for the authors

Experimental design

See comments in General comments for the authors

Validity of the findings

See comments in General comments for the authors

Additional comments

The manuscript of Dr. Maria Rakka et al. deals with a deep-sea octocoral and emphasizes the little knowledge existing on their early life history stages. The study describes the embryo and larval biology of Dentomuricea aff. meteor which is common species in the Azores. The experimental work was conducted under two temperature regimes, corresponding to the minima and maxima existing in their natural habitat during the reproductive season.
See comments below:
1. Indeed, the life history and reproductive biology of deep sea octocorals is less known compare to the shallow-water species. In this respect the body of information presented in the manuscript is novel and interesting. However, there are some flaws in the narrative of the manuscript which are addressed below.
2. The authors present the goal of the text (lines 83-88), but I would rather expect reading a clear working hypothesis with some specific research questions. Otherwise, the current narrative weakens the text and mainly the interpretation of the results. In its current format the manuscript is a kind of a descriptive report which can be acceptable, but I think it does not fit Peer J. As correctly stated by the authors, temperature is a critical environmental factor in the marine environment which greatly affects coral reproduction. The most intriguing question that comes up while reading the ms is what are the effects of ambient seasonal temperature changes on the early developmental stages and life history of D. aff. meteor. However, it is not clear what is the relevance of global change, in particular global warming to the present study, since it deals with natural and normal temp. fluctuations in the deep sea environment and do not present evidence for temp. change there.
3. A general comment: throughout the text where average± SD values are presented, sample size (n value) should be given (e.g. line 27, 29, 264, 270, etc.)
4. Lines 102-107 describe the conditions in which the octocoral fragments were maintained and indicate that the seawater was pumped from 5 m. Has the water quality in terms of pH, nutrients, salinity, etc. been compared to the seamount ones? If I read correctly, only the temperature was well controlled. Similarly, it sees that the filtration and UV treatments were applied in order to remove any microorganisms from the water and making them sterile. Did the author use any control to test possible effect of these procedures on the octocoral fragments?
5. The materials and methods section dealing with the embryonic development, larval studies, and their settlement and metamorphosis. I couldn’t find an indication which type of water was used for those assays. Did the authors use the same filtered/UV treated seawater as for rearing the fragments?
6. The Results section that deals with the spawning activity (line 229 onward) mentions that it has not been possible to directly observe fertilization and to differentiate between the possibility of internal vs. external fertilization. It should be noted that SEM examination, could have answer this question by looking for sperm in the polyp cavities or the oocyte surface,
7. I have some doubts concerning the results presented in the settlement and metamorphosis section (line 303 onward). The authors did not indicate if the basalt substrate was sterilized or had some organic film on it. Since it is well established in the literature that organic film may induce settlement and metamorphosis, this parameter should be considered while running those experiments. Similarly, there is no indication for the substrate texture and whether it played any role in enhancing settlement. Indeed, settlement took place also on the glass but the settlement rate must be normalized per substrate-type before drawing conclusions.
8. Figs. 2 & 4 present percentages and Fig. 3 presents probabilities, but sample size (n values) are missing which makes it hard to consider the strength of the results.
9. The discussion seems to me too long and some paragraphs are wordy (e.g. line 433 onward). For the sake of clarity, I recommend to present a comparative table with the different reproductive/life history traits of the studied species in comparison to previously studied deep sea octocorals, or even some shallow water ones. For the latter, I wonder why the authors did not cite the only existing review paper on octocoral reproduction by Kahng et al. 2011.
10. The Conclusion section (line 480 onward) should be rewritten to reflect both the novelty of the results and their general significance. For example, the sentence (line 489): “However, species distribution is a result of complex interactions between factors in various ecological levels” includes two vague notions (in bold). Those two should be clearly explained.


.

---

## Round 0.2 · Minor Revisions

I have received the final evaluation of your manuscript and there are a number of minor issues to attend to in a revision.

·

Basic reporting

no comment

Experimental design

no comment

Validity of the findings

no comment

Additional comments

The manuscript has been appreciably improved. I am satisfied with authors' corrections and answers. However, before publication, the authors would need to still address some little corrections. I also include some suggestions that I hope can be useful. Finally, I would like to congratulate the authors for the magnific work done. Working with coral larvae is not easy and, sometimes, it is very stressful. Good job!

---

## Round 0.3 · accepted · Accept

I am satisfied with the changes made to the manuscript.